# Social Stigma during COVID-19 and its Impact on HCWs Outcomes

**Tiziana Ramaci** [1], **Massimiliano Barattucci** [2,*], **Caterina Ledda** [3] **and Venerando Rapisarda** [3]

1   Faculty of Human and Social Sciences, Kore University of Enna, 94100 Enna, Italy; tiziana.ramaci@unikore.it
2   Faculty of Psychology, e-Campus University, 22060 Novedrate (CO), Italy
3   Occupational Medicine, Department of Clinical and Experimental Medicine, University of Catania, 95121 Catania, Italy; cledda@unict.it (C.L.); vrapisarda@unict.it (V.R.)
*   Correspondence: massimiliano.barattucci@uniecampus.it

**Abstract:** The COVID-19 emergency has significantly transformed the working environment and job demands. Providing care was emotionally difficult for healthcare workers. Uncertainty, stigmatisation, and potentially exposing their families to infection were prominent themes for healthcare workers (HCWs) during the crisis, which first broke out in China at the end of 2019, and then in Italy in early 2020. This study examined the effects of stigma, job demands, and self-esteem, and the consequences of working as a "frontline care provider" with patients infected with the coronavirus (COVID-19). A correlational design study involved 260 healthcare workers (HCWs) working in a large hospital in southern Italy. The following questionnaires were administered: (1) the Job Content Questionnaire (JCQ), for assessing psychological and physical demands; (2) the Professional Quality of Life Scale (ProQOL) to measure the quality individuals feel in relation to their work as "frontline care providers", through three dimensions: compassion fatigue (CF), burnout (BO), and compassion satisfaction (CS); (3) the Rosenberg Self-Esteem Scale, for evaluating individual self-esteem; (4) a self-administered multiple-choice questionnaire developed by See et al. about attitudes of discrimination, acceptance, and fear towards HCWs exposed to COVID-19. The findings suggest that stigma has a high impact on workers' outcomes. Stigma may influence worker compliance and can guide management communication strategies relating to pandemic risk for HCWs.

**Keywords:** COVID-19; stigma; job demands; self-esteem; fatigue; burnout; satisfaction; HCWs

## 1. Introduction

The COVID-19 pandemic first struck Italy in January 2020, when two Chinese tourists tested positive for SARS-CoV-2 in Rome. An outbreak of subsequent infections was subsequently detected, beginning with 16 confirmed cases in Lombardy on 21 February, rising to 60 cases the following day, with the first deaths were reported. At the time of writing, there are over one million 300 thousand people infected with the coronavirus worldwide and the number of deaths stands at almost 75 thousand, almost 85% of which have been registered in Italy, Spain, France, and the United Kingdom [1].

The pandemic crisis has significantly transformed the working environment and job demands (e.g., high-pressure work, an unfavorable physical environment, and emotionally demanding interactions). Providing care was emotionally difficult for healthcare workers, with stress, uncertainty, and stigmatisation being dominant themes for healthcare workers (HCWs). They often had complex and conflicting thoughts and feelings about balancing their roles as healthcare providers and parents, feeling professional responsibility but also fear of this new disease, associated coronavirus patients, and guilt about potentially exposing their families to infection by working during the COVID-19 emergency [2–5]. Working with potentially highly infectious patients led to considerable

stigmatisation [6,7]. Contagion brings out a whole range of attitudes, beliefs, prejudices, stereotypes, and stigmas. Under these conditions, emotions play a key role by distorting planned choices or those based on facts. There is a contradiction between the duty owed by doctors, nurses, and healthcare workers to their patients and the underlying attitudes caused by the contagion. In some cases, this can lead to prejudice against those who are seen as modern day "plague spreaders". The overriding fear is that of becoming infected, making the management of contact with infected individuals or those waiting for diagnostic test results difficult [8].

One of the most typical reactions in these cases is to experience fear, a primary emotion, which is crucial to our self-defence and survival. It is this fear that can lead healthcare workers to provide treatment that is less precise or careful than that which they would provide under normal circumstances [9].

The implications of working with potentially highly infectious patients should be recognised and acknowledged.

In this context, it is therefore essential to understand the effects of stigma, related to the intensity and frequency of exposure to the ongoing pandemic, job demands, and self-esteem, and its impact on HCWs outcomes. In particular, it is essential to investigate whether these variables are potentially capable of producing changes to the quality of professional life, including compassion satisfaction, burnout, and compassion fatigue for HCWS; in addition, it is also possible to hypothesise on the contribution of contextual variables, such as organisational type, position, years of experience, and role.

Stigma and discrimination tend to persist in the long term, even after quarantine has ended and the epidemic has been contained. HRM (Human Resourced Management) can positively support efforts to reduce stigma among HCWs and the related stress generated by increased workloads and being assigned to unfamiliar tasks. Both systematic training and specific network meetings, as well as the possibility to access counselling seem to be very important tools to fight burnout and social stigmas [10].

## 2. Social Stigma with Coronavirus Disease Patients

Stigma can be defined as a mark of disgrace that sets a person apart from others [11]. Social stigma (e.g., discrimination and devaluation by others) has a variety of negative consequences that inhibit recovery, such as shame, embarrassment, and the "why try" phenomenon [12,13].

Social stigma, in the context of health, is the negative association related to people or a group who have a specific disease in common. In an epidemic, this may mean that people are labelled, stereotyped, and discriminated against because of a perceived link to the epidemic. This is even more true when dealing with a highly contagious disease. This can have a negative effect on those affected by the virus and on the work of HCWs [14,15].

Firstly, stigmatisation can substantially increase the suffering of people with the disease. Secondly, people with the disease or those at risk of catching it may avoid seeking health care, making it much harder for public health authorities to control the disease. Thirdly, professionals and volunteers working in the field may also become stigmatised, leading to higher rates of stress and burnout [16–20].

Familiarity (e.g., knowing a friend or family member who has tested positive) is well established as a factor that positively impacts stigma [21]. Specifically, familiarity has been associated with lower levels of perceived dangerousness and fear [22] and less desire for social distance [23,24], as well as increased sympathy and prosocial attitudes [25].

Discrimination towards patients is the behavioural response of prejudice [26,27] and can be understood in terms of social processes of power and domination with some groups, which serve to devalue the stigmatised [28].

Evidence clearly shows that stigma and fear of infectious diseases hinder HCWs of different roles and responsibilities from responding correctly. They are facing an unprecedented emergency and insidious invisible danger, which has pushed the national health service to its limits, increasing workloads and

physical and mental stress. At the individual level, stigma has been associated with insufficient levels of knowledge [29] and fear of casual transmission in the workplace [30,31].

Further examination of the factors relating to stigma has resulted in associations between stress and satisfaction [32,33].

For all these reasons, the acute stress of working with potentially highly infectious patients should be recognised and acknowledged.

A number of models have been put forward by the literature for the study of workplace health that investigate the relationship between stress perceived by the worker [34] and available resources. The starting point for these studies is the perceived level of stress, seen as a result of an imbalance between the demands imposed by the situation and the individual personal resources available [35].

Individual resources are also very important in protecting HCWs against the negative effects of infection [36]. In fact, O'Keefe [37] found that strong self-esteem was the strongest predictor of hopefulness among HCWs with patients affected by virus [38]. Additionally, [39–41] suggested that after an individual experienced adversity, a higher sense of self-esteem was identified as one of the personal characteristics contributing to resilient psychosocial outcomes. On the contrary, low self-esteem may be a risk factor contributing to negative psychological outcomes [42,43].

Less attention has been paid by researchers to the pandemic situation and life satisfaction, and how these may impact on attitudes toward HCWs. Stigma-related stress is not a diagnosable concern, but it can lead to more serious direct consequences for workers' outcomes and their performance [44]. It could be that when workers experience increased stigma-related stress, they feel more inclined to assist with patients' health concerns. The opposite may be true for those experiencing high levels of stigma-related stress, where stigma may inhibit an individual from providing treatment. Similarly, satisfaction with life may be inversely related—with treatment provided and good performance outcomes when the individual feels satisfied with their current professional life circumstances, and perhaps more likely to provide support when satisfaction levels are higher. In a study with counselling professionals [45,46], help counsellors who reported higher self-stigma also had less help behaviours. This lack of behaviours then contributed to higher levels of stress and burnout and lower satisfaction.

In general, people who had higher levels of stigma were less satisfied. This finding suggests that when a person feels stressed, levels of satisfaction decrease [45,47–49].

Another study found that doctors who carried out abortions faced significant workplace stigma, resulting in reluctance due to workplace strain [50].

In such a context, an increase in job demands (e.g., psychological overload) exposes the individual to a tangible risk of burnout with cognitive, behavioural, emotional, and physical consequences, such as tiredness, pervasive detachment from others, anxiety, irritability, insomnia, poor concentration and indecision, degradation of performance levels, and reluctance to carry out one's work [51–53].

Literature on psychological consequences of exposure to the COVID-19 emergency reported emotional strain, burnout, and physical symptoms, such as shortness of breath and headaches, which were attributed to continually wearing protective masks, while fear and anxiety associated with the risk of contracting COVID-19 was prominent in their minds [54–57]. Authors found that although healthcare workers carried out their duties, the dual role of healthcare worker and family member caused conflict. Respondents were particularly concerned about infecting family and friends who they considered vulnerable. Other studies on the emergency found that HCWs were worried about expected overtime hours if other staff were quarantined, as well as the stigma of the illness and the health of their families and themselves [58], indicating general emotional distress [59–61]. Several studies have investigated the attitudes, knowledge, and practices of healthcare workers (HCWs) towards patients with the virus and underlined that HCWs still fear the disease and behave prejudicially toward infected patients [62–64]. Factors that influence these attitudes include fear of contagion associated with the uncertainty of care and the awareness of feeling useless in providing care for patients with a potentially fatal disease [65]. The focus of institutions and the scientific community on occupational health and safety is progressively increasing, leading to a continuous regulatory evolution and the development

of good practices in safety and prevention. Easily accessible practical advice on coping strategies and stress management at work may be a most challenging task and useful to significantly improve and guarantee their quality of life and work, and to avoid burnout.

## 3. Study Aim and Hypotheses

Therefore, after considering these research findings, the specific objective of the present preliminary analysis was to identify direct and indirect relationships between stigma, job demand, and quality of professional life, including compassion satisfaction, burnout, and compassion fatigue, in a group of HCWs working in a large hospital in the south of Italy with a COVID-19 ward.

In this hypothesis-generating study supported by a convenience sample drawn in close temporal proximity to the period of lockdown, imposed on the entire country by the government in an attempt to flatten the curve the pandemic (range time from 17 March to 2 April), we examined whether the relationships between these constructs existed for workers and to what extent. The data are still being constantly updated to provide additional support for the model presented in this paper.

In summary, according to the literature it is clear that stigma influences work outcomes [45]. Moreover, some working environment variables (perceived job demands) [35,51] and personal variables (self-efficacy) [38,42] could have a role in possibly mediating/moderating stigma and outcomes. Self-efficacy could influence the perception of stigma, increasing discrimination and fear of COVID-19 (Figure 1). However, this study was only intended to verify the first step of the theoretical framework. Stigma, job demands, and self-efficacy have the role of antecedents in relation to the outcomes of HCWs, overall making a joint contribution to the experience of work.

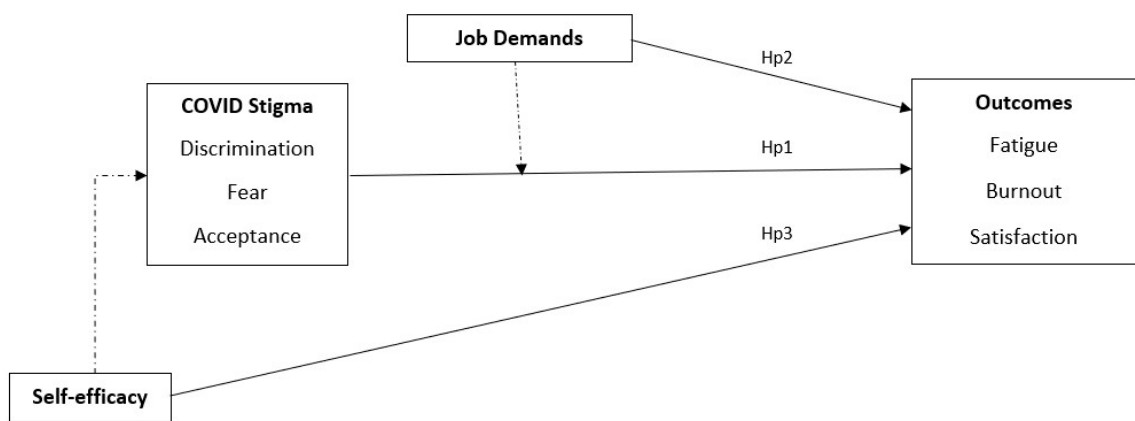

**Figure 1.** Overall theoretical model of stigma at work.

Based on this simple rationale, and with a view to further exploratory research, this paper intended to verify the following hypotheses:

**Hypothesis 1 (H1)**: *Social stigma (discrimination, acceptation, and fear) predicts outcomes: stigma discrimination, and fear negatively predict compassion fatigue (CF) and burnout (BO) (H1a), and positively predict ompassion satisfaction (CS) (H1b). On the contrary, stigma acceptance positively predicts fatigue and burnout (H1c), and negatively predicts satisfaction (H1d).*

**Hypothesis 2 (H2)**: *Job demands (mental and physical overload) predict outcomes, such as professional quality of life. More specifically, job demands positively predict compassion fatigue (CF) and burnout (BO) (H2a), and negatively predict compassion satisfaction (CS) (H2b).*

**Hypothesis 3 (H3)**: *Self-esteem negatively predicts negative outcomes (fatigue and burnout) and positively predicts satisfaction levels.*

Other contextual variables such as gender, age, role, length of service, and working hours were considered as control variables.

## 4. Materials and Methods

### 4.1. Sample

A cross-sectional study was carried out from the 17th to the 26th of March 2020, with the healthcare workforce from the National Health Service hospitals in Sicily, who took part in the study on a voluntary basis. The procedure required researchers to administer questionnaires during working hours, using an accessibility sampling.

The study related to this paper was carried out in accordance with APA ethical standards. In line with the ethical standards of the 1964 Helsinki Declaration, before taking part in the study, participants were informed about all relevant aspects of the study (e.g., methods, institutional affiliations of the researcher). They were informed of their right to refuse to participate in the study or to withdraw consent to participation at any time during the study without reprisal. They then confirmed that they fully understood the instructions, verbally accepted to take part, and began to complete the anonymous questionnaire.

The final sample was made up of 273 health care workers from a university hospital (mean age = 46.67, SD = 8.36; 137 women, 136 men), 67 nurses (response rate on the total sample of nurses = 22%; 24.5%), and 206 doctors (response rate on the total sample of doctors = 94%; 75.5%). Average organisational tenure was 13.32 years (SD = 10.7). Overall, 38% of the subjects were fixed term (N = 104), 33.5% were unmarried (N = 91), and 46.6% had no children (N = 127).

### 4.2. Measures

This study was conducted using the following self-administered multiple-choice questionnaires:

The Professional Quality of Life Scale (ProQOL) developed by Stamm [66,67] aims to measure the professional quality of life of accident and emergency workers based on three dimensions: assessment of risk of compassion fatigue (CF), potential for compassion satisfaction (CS), and risk of burnout (BO). Higher scores on the compassion fatigue subscale (C: 7 items) indicate the respondent is at higher risk of compassion fatigue (e.g., "I felt satisfied with my ability to cope with emergency procedures and techniques"). Higher scores on the compassion satisfaction subscale (CS: 8 items) indicate the respondent is experiencing higher satisfaction with their ability to provide care (i.e., caregiving is an energy-enhancing experience, increased self-efficacy) (e.g., "I was startled or agitated when I heard sudden noises"). Higher scores on the burnout subscale (BO: 7 items) indicate the individual is at risk of experiencing symptoms of burnout (e.g., "I felt I was experiencing the same trauma as the person I was treating"). Items are rated on a 5-point response scale, ranging from 1 = never, 2 = rarely, 3 = sometimes, 4 = often, and 5 = very often. The coefficient alpha for compassion satisfaction was 0.9, 0.82 for compassion fatigue, and 0.82 for burnout.

The Job Content Questionnaire (JCQ) by R.A. Karasek [68,69]. The JCQ is a self-administered instrument designed to measure social and psychological characteristics of jobs, the best-known scales—(a) decision latitude, (b) psychological demands, and (c) social support (49 items)—are used to measure the high-demand/low control/low-support model of job strain development, to assess job strain. The demand/control model predicts, first, stress-related risk and, second, active–passive behavioural correlates of jobs. For the purposes of this study, the following was investigated: the high work pressure demands (3 items) relating to mental load (resulting from lack of clarity and heavy time pressures faced by the individual in their job, e.g., "My job requires me to do things very quickly", "My job is very mentally demanding") (the coefficient alpha was 0.69) and unfavourable demands of the physical environment (1 item), relating to the physical effort required by an individual's job (e.g., "My job involves intense physical effort"). Items are rated on a 4-point response scale, ranging from: 1 = definitely no, 2 = no, 3 = yes, 4 = definitely yes.

The Rosenberg Self-Esteem Scale [70,71]. A 10-item scale that measures global self-worth by measuring both positive and negative feelings about the self. The scale is believed to be unidimensional (the coefficient alpha was 0.92) (e.g., "I feel that I have a number of good qualities"). All items are answered using a 4-point Likert scale format ranging from strongly agree to strongly disagree.

A self-administered multiple-choice questionnaire originally developed by See et al. [72] (adjusted) was administered on attitudes of discrimination, acceptance, and fear towards these HCWs exposed to COVID-19. This tool allowed us to evaluate three aspects: discrimination, acceptance of COVID-19 patients, and fear (the coefficient alpha for discrimination was 0.83, 0.56 for acceptance, and 0.72 for fear). For each of the three aspects there were four multiple choice questions with answers ranging from: strongly disagree = 0, disagree = 1, agree = 2, strongly agree = 3. Negative questions were reverse-scored to ensure the direction was consistent for all items and higher scores represented a more positive professional attitude. The original version of the questionnaire was translated into Italian by a professional mother tongue translator. Due to the Cronbach's alpha value of the acceptance sub-scale, this measure was excluded by data analysis.

Socio-demographic variables—participants were asked to provide information on socio-demographic characteristics, such as gender, age, and job-related variables, such as role, position, job shift, length of service, and hours worked.

### 4.3. Data Analysis

The research design was correlational. Due to the exploratory nature of the study, the research investigated relationships between variables (antecedents and outcomes) with correlation analysis and multiple regressions. Furthermore, in order to look for differences in the measured variables related to socio-demographical variables, a series of independent sample t-tests, an ANOVA, and a correlational analysis were carried out. Differences in the temporal trend of the variables were calculated taking into account the day of compilation (day 1 to day 10 of administration).

## 5. Results

Gender differences emerged for fatigue ($t_{273}$ = −2.735; $p < 0.01$) and burnout ($t_{273}$ = −3.087; $p < 0.001$): women reported higher scores of compassion fatigue and burnout than men (Table 1). Age was significantly positively related only to burnout levels (0.206**; $p < 0.01$) and satisfaction (0.189*; $p < 0.05$), as was length of service (burnout = 0.204**; $p < 0.01$; satisfaction = 0.202**; $p < 0.01$).

**Table 1.** Gender differences among the variables of the study.

|  | Female (N = 137) M (*SD*) | Male (N = 136) M (*SD*) |
|---|---|---|
| **Stigma Discrimination** | 1.45 (*0.66*) | 1.57 (*0.70*) |
| **Stigma Fear** | 3.1 (*0.75*) | 3.07 (*0.71*) |
| **Satisfaction** | 3.56 (*0.97*) | 3.81 (*1.0*) |
| **Psych. Job Demand** | 3.04 (*0.63*) | 3.09 (*0.66*) |
| **Self-efficacy** | 1.09 (*0.75*) | 1.07 (*0.92*) |
| **Fatigue** | 2.53 (*1.2*) | 2.13 (*1.3*) ** |
| **Burnout** | 2.02 (*0.99*) | 1.62 (*1.0*) *** |

*** $p < 0.001$; ** $p < 0.01$, * $p < 0.05$.

No profile differences (doctors vs. nurses) were found for any of the measured variables, nor for shift presence/absence. Furthermore, weekly working hours were not significantly related to any of the variables. Nevertheless, differences between temporary and long-term workers emerged for job demands ($t_{273}$ = −2.035; $p < 0.05$), fatigue ($t_{273}$ = −2.077; $p < 0.05$), and burnout ($t_{273}$ = −3.434;

$p < 0.001$)—unexpectedly, permanent workers showed higher levels of perceived psychological job demands, fatigue, and burnout, compared with temporary workers (Table 2).

**Table 2.** Differences between workers with different contracts among the variables of the study.

| | Fixed-Term (N = 84) M (*SD*) | Long-Term (N = 137) M (*SD*) |
|---|---|---|
| **Stigma Discrimination** | 1.42 (*1.08*) | 1.57 (*0.94*) |
| **Stigma Fear** | 3.1 (*0.70*) | 3.06 (*0.75*) |
| **Satisfaction** | 3.62 (*1.1*) | 3.72 (*0.94*) |
| **Psych. Job Demand** | 2.8 (*0.66*) | 3.2 (*0.63*) * |
| **Self-efficacy** | 1.05 (*0.84*) | 1.09 (*0.85*) |
| **Fatigue** | 2.11 (*1.15*) | 2.48 (*1.29*) * |
| **Burnout** | 1.57 (*0.99*) | 1.98 (*0.97*) ** |

*** $p < 0.001$; ** $p < 0.01$, * $p < 0.05$.

Table 3 presents descriptive statistics and zero-order correlations among the measured variables. In order to check for the hypothesised relationships between variables, we performed a multiple regression with stigma, job demand, and self-efficacy as predictors of outcomes. Results are presented in Table 4.

**Table 3.** Descriptive statistics and zero-order correlations among the variables of the study.

| | M (*SD*) | 1 | 2 | 3 | 4 | 5 | 6 | 7 |
|---|---|---|---|---|---|---|---|---|
| 1. Quarantine Day | 5.37 (*2.74*) | | | | | | | |
| 2. Stigma Discrimination | 1.57 (*0.72*) | 0.109 | | | | | | |
| 3. Stigma Fear | 2.88 (*0.89*) | −0.452 *** | 0.159 ** | | | | | |
| 4. Psych. Job Demand | 3.06 (*0.66*) | 0.036 | −0.025 | 0.021 | | | | |
| 5. Self-efficacy | 1.09 (*0.85*) | 0.026 | 0.160 ** | 0.064 | 0.093 | | | |
| 6. Satisfaction | 3.68 (*1.1*) | 0.121 * | −0.263 ** | −0.173 ** | 0.083 | −0.187** | | |
| 7. Fatigue | 2.22 (*1.3*) | −0.173 ** | 0.307 *** | 0.341 *** | 0.199 ** | 0.185 ** | −0.366 *** | |
| 8. Burnout | 1.82 (*0.97*) | 0.028 | 0.348 *** | 0.226 ** | 0.144 * | 0.113 | −0.178 ** | 0.745 *** |

*** $p < 0.001$; ** $p < 0.01$, * $p < 0.05$.

**Table 4.** Multiple regression of stigma, job demand, and self-efficacy as predictors of outcomes.

| Variables | Satisfaction | | | Burnout | | | Fatigue | | |
|---|---|---|---|---|---|---|---|---|---|
| | B | Beta | t | B | Beta | t | B | Beta | t |
| Stigma Discrimination | −0.318 | −0.231 | −3.73 *** | 0.424 | 0.317 | 5.54 *** | 0.431 | 0.248 | 4.5 *** |
| Stigma Fear | −0.175 | −0.141 | −2.54 ** | 0.186 | 0.171 | 3.08 ** | 0.413 | 0.291 | 5.41 *** |
| Psych. Job Demand | 0.124 | 0.082 | 1.38 | 0.175 | 0.125 | 2.27 * | 0.345 | 0.187 | 3.33 *** |
| Self-efficacy | −0.175 | −0.146 | −2.35 * | 0.048 | 0.042 | 0.736 | 0.183 | 0.125 | 2.1 * |
| $R^2$ | 0.13 *** | | | 0.19 *** | | | 0.24 *** | | |

*** $p < 0.001$; ** $p < 0.01$, * $p < 0.05$.

Among correlational results, COVID-19 fear levels and fatigue in HCWs decreased over the temporal trend, while satisfaction slightly increased.

Globally, correlations and regression analysis confirmed all the hypothesised relationships between variables. More precisely, stigma positively predicts burnout and fatigue (Hp1a) and negatively predicts satisfaction (Hp1b). On the contrary, job demands only predict negative outcomes (Hp2a), while self-efficacy slightly predicts two of the outcomes (fatigue—Hp3a, and satisfaction—Hp3b).

Overall, results showed that of the antecedents, stigma discrimination and fear are strong predictors of outcomes. All antecedents, in particular, significantly predicted negative outcomes: fatigue for 24% of variance, burnout for 19%.

## 6. Discussion

Measuring the effect of pandemic factor stigma on workers' performance is of extreme importance [55,56,73]. To this end, the research sought to provide preliminary indications on the relationship between stigma and work outcomes, and on the role of job demands and self-efficacy. The results undoubtedly show that stigma positively impacts fatigue and burnout, and negatively impacts satisfaction. The role of job demands, although having an effect on negative outcomes, appears to be reduced compared to the interaction with stigma perceptions. Self-efficacy also appears to relate more to the processes of discrimination and satisfaction than to those of emotional reaction (fear) and negative outcomes.

Stigma is such a pressing issue for the national health system, it has been identified as a health crisis that clinicians must take action against [74]. HCW stigmatisation is associated with psychological and physical health. HCWs who expected to experience higher levels of stigmatisation reported increased psychological distress, and this predicted increased somatic symptoms [75].

There are some major pathways for studying stigma in healthcare facilities, namely stigma related to discrimination and fear of contracting the virus and its outcomes [76–78]. Where HCWs are not aware of potentially stigmatising attitudes and behaviours, the impact of stigma is serious. The practical reason for exploring stigmatised attitudes and behaviours, and reducing related stigma, is the negative effect stigma has on a person's self-concept [79,80], life satisfaction [81,82], and professional quality of life, stress, burnout, and self-engagement [81,83].

It is no surprise then, that stigma toward HCWs has been a topic of focus in the literature [26,84–87].

There are several potential mechanisms by which stigma could affect HCWs outcomes [88,89]. Many research studies have been conducted to study the ways in which stigma impacts help behaviours [90–92]. The importance of stigma to quality of life (QOL)is well-recognised in HIV research and care: Stigma is included as a domain in the World Health Organisation's HIV-specific measure of QOL [93]. Caring for people living with a virus requires ongoing health care services, as they are potentially at increased risk of developing disorders, including cardiovascular and liver disease, accelerated bone loss, metabolic disorders, etc. [94,95]. Taking care of infected patients requires HCWs to have good knowledge of their unique issues. Cultural differences in HCWs, combined with professional ethics and personal beliefs, could also result in conflicting attitudes, which may lead to difficulties related to care [65,96]. Although most workers rationalised this as a lack of understanding about the illness or the risks involved, all described feeling angry and hurt, acutely aware of others' reactions.

Overall, on one hand, the results of this research seem to provide indications in line with cited literature and with the proposed theoretical model (Figure 1), but on the other, the range of relationships and the sample size do not allow for causal inferences or hasty conclusions to be drawn.

Indeed, the limited size of the sample can only provide preliminary indications and does not allow results to be generalised for all HCWs. Moreover, the very low response rate of the nurses was certainly caused by the lower temporal availability compared to doctors, and it can certainly represent an important source of bias and a loss of important information.

The outbreak of COVID-19 in Italy is a unique historical event that will need to be investigated more extensively and with more refined methodologies [54–56].

What is certain is that it is essential to study workers' stigma in the face of pandemics and the training and information provided for HCWs to ensure adequate levels of satisfaction can be maintained and prevent phenomena such as fatigue and burnout.

Research relating to the set of different antecedents of workers' outcomes in pandemics seems crucial since stigma risk may influence the general compliance of workers and results can provide useful information for management communication strategies.

## 7. Conclusions

There is now a greater focus than ever on studying stigma in relation to healthcare workers. Where HCWs are not aware of potentially stigmatising attitudes and behaviours, the impact of stigma is serious. HCWs who expected to experience higher levels of stigma reported increased psychological distress, stressors which may be important in predicting impact on HCWs' outcomes [75–77]. Working with potentially highly infectious patients generates considerable stigmatisation [6,7].

Our findings underline that stigma is an important predictor of compassion satisfaction, burnout, and compassion fatigue among HCWs. Therefore, strengthening human resources for frontline care providers requires measures to reduce stigma.

This appears particularly relevant for HCWs in this specific situation, whose contact with patients during the COVID-19 emergency is emotionally difficult and where stigma can jeopardise outcomes and affect work performance. [2–5,54,55]. In line with the broader literature, our findings also suggest that studying the stigmatisation of COVID-19 may provide us with insight into the stigma associated with emerging infectious diseases and the potential consequences of such stigmatisation.

In the specific case of healthcare workers, coming into contact with patients is an emotional stressor that can pose a threat to well-being outcomes and have an impact on quality of professional life.

HRM can positively support efforts to reduce the job stress that is generated by increased workload and assignment to unfamiliar tasks. Systematic training and specific network meetings, as well as the possibility to access counselling, are very important tools to fight burnout and social stigma [10] in order to prevent them or avoid their harmful effects.

Despite the contribution made by this study to the understanding of the topic, there are limits which provide direction for future research. Firstly, the methods used to examine "causal" hypotheses and data collected were cross-sectional and, therefore, cannot offer evidence of actual causation. In future research, using a structural equation longitudinal method would be useful. Secondly, self-reported measures were used to assess the dimensions of this study. Future studies should at least consider different methods to reduce the influence of self-report bias. In this hypothesis-generating study carried out in close temporal proximity to the lockdown period, imposed by the government to attempt to flatten the curve of the pandemic, we used a convenient sample. The data are still being constantly updated to provide additional support for the model presented in this paper.

**Author Contributions:** Conceptualization, T.R. and M.B.; methodology, M.B. and T.R.; validation, T.R., M.B. and V.R.; formal analysis, M.B.; investigation, C.L. and V.R.; data curation, M.B.; writing—original draft preparation, T.R.; writing—review and editing, T.R. and M.B.; visualization, V.R. All authors have read and agreed to the published version of the manuscript.

**Funding:** This research received no external funding.

**Conflicts of Interest:** The authors declare no conflict of interest.

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
