# Peer review of "Social Stigma during COVID-19 and its Impact on HCWs Outcomes"

_sustainability, doi:10.3390/su12093834_

Round 1

Reviewer 1 Report

I think this article, described on the effects of COVID-19 pandemic on social stigma in health care workers, is very important in the current state of COVID-19 pandemic. I think it will be easier to read if the author modifies the configuration a little: “2. Social Stigma with coronavirus disease patients” is described in great detail. How about shortening this part and describing much of the content in the “Discussion”?

Reviewer 2 Report

I think this is an excellent paper and in strong shape. A couple of minor changes would help:

1) Please provide any available data to compare the study sample to the workforce of the institution. Does the sample reflect the organization or are there obvious sources of bias?

2) How was HCW defined? Does the sample include orderlies, nursing aides, administrative staff etc.....When the study concludes that there are no "profile" differences between doctors and nurses, does this also apply to lower-status staff or are they excluded from the study? I am thinking that folks in lower status jobs may experience the stresses of the pandemic differently.

3) In table 3, there is a variable "quarantine days".....and it seems that it is associated with fear, satisfaction and fatigue....I could not find a description of this variable....and if it remained significant in the multivariate models.

4) It is not clear whether or not the multivariate findings in Table 4 also includes controls for gender, age, and other factors found to be associated with the outcomes.

5) The study makes a couple of references to poor adherence/poor performance as an outcome of the lower rates of satisfaction, burnout and fatigue experienced by many of the sample members-----and I am not clear how this adherence/poor performance is actually measured in the study or if it is an expected product of the measured outcomes.

6) The paper needs one more read through for English syntax and usage....there are missing articles in some sentences and the use of abbreviations ("var") that need to be corrected.
